# The Relative Preservation of the Central Retinal Layers in Leber Hereditary Optic Neuropathy

**DOI:** 10.3390/jcm11206045

**Published:** 2022-10-13

**Authors:** Sanja Petrovic Pajic, Luka Lapajne, Bor Vratanar, Ana Fakin, Martina Jarc-Vidmar, Maja Sustar Habjan, Marija Volk, Ales Maver, Borut Peterlin, Marko Hawlina

**Affiliations:** 1Eye Hospital, University Medical Centre Ljubljana, Grablovičeva ulica 46, 1000 Ljubljana, Slovenia; 2Clinic for Eye Diseases, University Clinical Centre of Serbia, Pasterova 2, 11000 Belgrade, Serbia; 3Institute for Biostatistics and Medical Informatics, Faculty of Medicine, University of Ljubljana, 1000 Ljubljana, Slovenia; 4Clinical Institute of Genomic Medicine, University Medical Centre Ljubljana, Šlajmajerjeva ulica 4, 1000 Ljubljana, Slovenia

**Keywords:** ganglion cell complex, optical coherence tomography, retinal layer segmentation, peripapillary RNFL, Leber hereditary optic neuropathy

## Abstract

(1) Background: The purpose of this study was to evaluate the thickness of retinal layers in Leber hereditary optic neuropathy (LHON) in the atrophic stage compared with presumably inherited bilateral optic neuropathy of unknown cause with the aim of seeing if any LHON-specific patterns exist. (2) Methods: 14 patients (24 eyes) with genetically confirmed LHON (LHON group) were compared with 13 patients (23 eyes) with negative genetic testing results (mtDNA + WES) and without identified etiology of bilateral optic atrophy (nonLHON group). Segmentation analysis of retinal layers in the macula and peripapillary RNFL (pRNFL) measurements was performed using Heidelberg Engineering Spectralis SD-OCT. (3) Results: In the LHON group, the thickness of ganglion cell complex (GCC) (retinal nerve fiber layer (RNFL)—ganglion cell layer (GCL)—inner plexiform layer (IPL)) in the central ETDRS (Early Treatment Diabetic Retinopathy Study) circle was significantly higher than in the nonLHON group (*p* < 0.001). In all other ETDRS fields, GCC was thinner in the LHON group. The peripapillary RNFL (pRNFL) was significantly thinner in the LHON group in the temporal superior region (*p* = 0.001). Longitudinal analysis of our cohort during the follow-up time showed a tendency of thickening of the RNFL, GCL, and IPL in the LHON group in the central circle, as well as a small recovery of the pRNFL in the temporal region, which corresponds to the observed central macular thickening. (4) Conclusions: In LHON, the retinal ganglion cell complex thickness (RNFL-GCL-IPL) appears to be relatively preserved in the central ETDRS circle compared to nonLHON optic neuropathies in the chronic phase. Our findings may represent novel biomarkers as well as a structural basis for possible recovery in some patients with LHON.

## 1. Introduction

Leber hereditary optic neuropathy (LHON) is an inherited optic neuropathy that develops predominantly as a consequence of three mtDNA point pathogenic variants, namely m.3460G > A (MT-ND1), m.11778G > A (MT-ND4), and m.14484T > C (MT-ND6) [1]. Recently, LHON has also been reported due to autosomal recessive pathogenic variants in the DNAJC30 gene [2].

Due to high energy demand, retinal ganglion cells (RGCs) are particularly vulnerable to mitochondrial dysfunction, which may lead to apoptotic cell death and axonal degeneration, resulting in optic atrophy [3,4]. The papillomacular bundle consists predominantly of midget RGCs, which mediate visual information including high spatial frequencies and red/green chromaticity [4]. Being the axons with the smallest calibre axons, they have the least mitochondrial reserve per energy requirement. Therefore, these cells are particularly severely affected by the disturbed mitochondrial energy metabolism and increased formation of ROS that are the main biochemical characteristics of LHON [5,6,7].

Studies of retinal structure in vivo have shown that even in the pre-symptomatic stage of LHON, i.e., approximately 6 weeks before the onset of visual loss, thinning of the RGC-IPL is observed on the optical coherent tomography (OCT) in the fellow eye [8,9,10,11]. Severe thinning of the macular RGC-IPL layers continues for 3–6 months into the disease; later, the thinning process is no longer progressing [5,10]. On the other hand, peripapillary RNFL thickness initially increases and then decreases due to the retinal ganglion cell swelling and apoptosis. pRNFL thinning initially occurs in the temporal quadrant, then in the inferior and superior quadrants, and finally, in the nasal quadrant. The pRNFL thinning continues even after 60 months [9]. Up to 15% of LHON patients experience some degree of visual recovery [12,13]. One of the explanations for this improvement is the existence of the remaining non-apoptotic RGCs which enter a dysfunctional dormant state and can reactivate even after a long time [14,15].

Patients with typical LHON phenotype but without currently discoverable changes in either mitochondrial or nuclear DNA are a special optic atrophy subgroup and diagnostic enigma. They may have a similar phenotype to that of LHON, but genetic analysis shows no causative pathogenic variant.

The purpose of this study was to evaluate the thickness of retinal layers in the macular region as well as the peripapillary RNFL thickness in Leber hereditary optic neuropathy (LHON) in comparison to other primary, presumably inherited optic neuropathies without identified genetic cause in the atrophic stage with the aim to see if any specific patterns exist in LHON.

## 2. Materials and Methods

### 2.1. Patients

This study was conducted at the Eye Hospital, University Medical Centre Ljubljana, Slovenia. We initially identified 70 patients with chronic bilateral optic atrophy. The management protocol is presented in Figure 1. The exclusion criteria were as follows: all acquired causes of visual impairment (ischaemic, demyelinating, postinfectious, autoimmune, nutritional, or toxic optic neuropathies), and other known ophthalmic or neurological diseases that may lead to similar changes in the optic nerve. To identify such causes, all patients underwent head MRI and detailed diagnostic workup (complete screening tests for compressive, inflammatory, ischaemic, and autoimmune conditions, including anti-AQ4 antibodies, anti-MOG antibodies) and were referred to a neurologist for exclusion of any associated neurological disorders. All patients that remained in the study groups also underwent detailed genetic testing. After all exclusion criteria were applied, 30 patients remained in the study.

According to genetic testing results and phenotype findings, these patients were divided into two study groups. The first group, the LHON group, consisted of participants with a confirmed pathogenic variant in either mtDNA or nuclear DNA (nDNA), and phenotype consistent with LHON (14 patients, 24 eyes, mean age 37.5 years, 10 males).

The second group, the nonLHON group, consisted of participants in whom no pathogenic variants were found in either mtDNA or nDNA, but their phenotype included bilateral optic neuropathy without any other detectable cause (13 patients, 23 eyes, mean age 46 years, 7 males).

In three patients, dominant optic atrophy (DOA) was identified. This group consisted of 2 males, 60 years and 45 years, with OPA1 pathogenic variant, and a female of 40 years with ACO2-related optic neuropathy. As these three patients had different disease courses and results in comparison to both study groups, they were studied separately.

Finally, 24 eyes of 14 patients in the LHON group, and 23 eyes of 13 patients in nonLHON group were included in the comparative segmentation study. Due to amblyopia with strabismus and the inability to fixate, 4 eyes in the LHON group and 3 eyes in the nonLHON group were excluded from the analysis.

### 2.2. Genetic Analysis

All patients were tested at the Clinical Institute of Genomic Medicine, University Medical Centre Ljubljana, Ljubljana, Slovenia.

Patients were first tested for the three typical LHON pathogenic variants (m.3460G > A, m.11778G > A, m.14484T > C). Unless typical pathogenic variants were found, next-generation sequencing (NGS) of whole mitochondrial DNA (mtDNA) was done in search of atypical changes in mtDNA. In patients without detected changes in the mtDNA, a clinical and/or whole exome sequencing was also performed (Appendix A). According to the previously described protocol, next-generation sequencing of exome with mitochondrial genome-oriented testing, analysis, and filtration of found variants were performed on a DNA sample (28). Genetic analysis in “LHON group” revealed the following genotypes: 5 patients with MT-ND1: m.3460G > A, 3 patients with MT-ND4:m.11778G > A, 1 patient with MT-ND6: m14484 T > C, 2 patients with MT-ND5, m.13042G > T, 1 patient with MT-ND1:m.3700G > A, and two had autosomal recessive DNAJC30:c.152 A > G (p.Tyr51Cys) variant in the homozygous state (Appendix A). In all patients in “nonLHON” group, no pathogenic variants were found either in mtDNA or nDNA analysis.

Three patients with dominant optic atrophy were identified (DOA). This group consisted of: (i) male, 60 years, with OPA1: c.2635delC (p.Arg879GlufsTer3), class 5 pathogenic variant; (ii) male, 45 years, with OPA1: c.2489G > A, class 4 likely pathogenic variant; (iii) and female, 40 years, with ACO2:c.2253dupC in c.719G > C (Ile752HisfsTer13) (class 5 pathogenic variant) and in the same patient likely pathogenic ACO2:c.719G > C (Gly240Ala) (class 3 variant of unknown significance).

### 2.3. Ophthalmologic Examination and Workup

At each appointment, the diagnostic workup included full ophthalmological examination, best corrected visual acuity (Snellen), colour vision (Ishihara plates), visual field examination (Goldmann or Octopus perimetry) and spectral domain optical coherence tomography (SD-OCT). Fluorescein angiography (FA) was performed once, soon after presentation, and electrophysiology testing (pattern electroretinography-PERG, and visual evoked potentials-VEP) after presentation and at selected follow-up time points.

#### 2.3.1. Electrophysiology

Pattern ERG and VEP were performed according to the method described earlier [16]. Mean amplitudes and peak times of PERG P50 and N95 and VEP P100 wave were compared among the groups.

#### 2.3.2. Optical Coherence Tomography and Retinal Segmentation Study

The segmentation analysis data from the two study groups were compared with each other and with normative segmentation data obtained from 200 healthy controls using the same segmentation methodology (Spectralis OCT, Heidelberg Engineering, Heidelberg, Germany) [17].

At presentation and each successive follow-up visit, the participants underwent OCT imaging of the macular region and peripapillary RNFL. Using the methodology in the normative study [17] we studied retinal segmentation in the following ETDRS fields: central circle, middle ring (inner superior, inner inferior, inner temporal, and inner nasal field), and outer ring (outer superior, outer inferior, outer temporal, and outer nasal field) (Figure 2a). The data were analysed at different time points of the chronic phase at least 5 years after the onset of the disease.

The software automatically provided a thickness map by retinal layers in respective ETDRS fields in accordance with the consensus of the international nomenclature for OCT [18]. In addition, each image was carefully controlled. If the automatic setting was incorrect, inadequately defined layers and fixation errors were manually corrected. If the imaging quality was poor, measurements were repeated; only good-quality recordings were used, and Spectralis HRA + OCT software automatically calculated data for each sector. The GCC thickness was calculated by subtracting the measurements for RNFL, GCL and IPL layer.

#### 2.3.3. Peripapillary RNFL Thickness Study

The peripapillary RNFL thickness (pRNFL) in patients with LHON was also measured using Spectralis HRA + OCT. For pRNFL thickness measurement, a 3.5-mm-diameter circular scan, centered on the optic disc, was used and the data for six sectors (superior temporal (ST), inferior temporal (IT), temporal (T), superior nasal (SN), inferior nasal (IN) and nasal (N) fields) and the global 360° average value (G) were collected (Figure 2b).

### 2.4. Statistical Analysis

The thickness of every retinal layer in each ETDRS ring was calculated by averaging the values of corresponding ETDRS fields in each eye. These were also calculated separately.

In comparisons of both study groups with the control group [17], only one eye was selected, and the mean values and standard deviations for one eye were reported. For each patient, the right eye was selected; in case of amblyopic or poorly fixating right eye, left eye was used. The mean thickness of the retinal layers between LHON and healthy population as well as between nonLHON and healthy population was compared using Welch’s t-test for independent samples for each retinal layer [19]. For each layer, the ratio between the central circle and the other two ETDRS rings was calculated and mean values between LHON and nonLHON groups were compared using Welch’s t-test. To examine the relationship between two binary variables, Fisher’s exact test was used.

To increase statistical power of comparisons of differences in thicknesses of particular retinal layers between LHON and nonLHON group, the data from both eyes of each patient were used for this part of the study. Since the left and the right eye of the same patient are highly correlated ($r_{min}=$ 0.35, $r_{max}=$ 0.87, $\bar{r}$ = 0.7), the data were analysed using mixed models method [19] n such cases, this method is most appropriate, as standard statistical tests (e.g., t-test, linear regression) assume that the observations are independent. Two models were fitted to our data for each retinal layer and ETDRS field: In the first (baseline) model, only the random part of the model (participant’s ID) was included, and in the second (full) model, diagnostic group (LHON or nonLHON) as a fixed factor and participant’s ID as a random factor were included. The two models were compared using the Likelihood ratio test. The difference between the two models is the evidence against the null hypothesis, which states that the mean retinal thickness is the same between the two groups. Bonferroni–Holm correction was used to adjust *p*-values for multiple comparisons. For all statistical tests, the significance level was set at α = 0.05. All statistical analyses were performed with R version 3.5.1 (R Foundation, https://www.r-project.org/foundation/, accessed on 27 August 2022) [20].

## 3. Results

### 3.1. Patient Demographic Data

Patient demographic data are shown inAppendix A. The two patient groups had a similar phenotype that consisted of subacute bilateral painless visual loss, centrocecal scotoma, and atrophic optic nerve head in the chronic stage of the disease.

### 3.2. Electrophysiology

Electrophysiology results are presented in Appendix A. The N95 PERG wave was reduced in all recorded patients (22 of 27, in 5 patients electrophysiology was not performed), and there was no difference between LHON and nonLHON group in N95 amplitude (*p* = 0.662). In the LHON group, VEP P100 wave was delayed and decreased in all patients, in 7 patients being undetectable in the late chronic phase (63.63% median 42 months after onset), whilst in the nonLHON group only 2 patients (18.18%) had undetectable VEP (median 48 months after onset). Out of 4 patients with detectable VEP P 100 wave in the LHON group, 3 patients had gradual visual acuity and color vision improvement. There was no statistically significant difference between the LHON group and the nonLHON group regarding mean P 100 wave amplitude and peak time (*p* = 0.196 and *p* = 0.804). The relationship between the group and the number of patients with undetectable VEP was also not statistically significant (*p* = 0.081 and *p* = 0.149 for the right and the left eye, respectively. In addition, electrophysiology did not reveal significant differences in functional patterns between the groups.

### 3.3. Segmentation Analysis

#### 3.3.1. Comparison of Retinal Layers of LHON and nonLHON Group against Healthy Controls

Retinal thicknesses and mean differences between the control group versus LHON and the nonLHON group are shown in Figure 3 (values are presented in Appendix A).

Ganglion cell complex (GCC; ganglion cell layer (GCL) – inner plexiform layer (IPL)) in the LHON and nonLHON groups was significantly thinner than controls in the central circle, as well as in the middle and outer ring. On the contrary, in LHON, INL was thicker in the central circle and outer ring in comparison to controls (Table 1,Appendix A). The differences in mean thickness in the outer layers (outer plexiform layer (OPL) and outer nuclear layer (ONL)) are small and statistically insignificant (Figure 2 and Table 1).

#### 3.3.2. Longitudinal Analysis

Longitudinal analysis of our cohort during the follow-up time showed a trend of thickening of the GCC in the LHON group in a central circle (Figure 4).

A similar trend was not observed in the nonLHON group, where GCC remained the same or thinned slightly. The thickness of these layers in middle and outer ETDRS rings remained the same or thinned during the chronic phase in both groups. In Appendix A, the longitudinal data for other macular layers are shown.

#### 3.3.3. Comparison of Retinal Layers’ Thicknesses in ETDRS Rings between LHON and nonLHON Group

Using mixed models to include both eyes, in our analysis, we compared mean retinal thickness between LHON and nonLHON group for each combination of retinal layers and rings. In the LHON group, all retinal layers are thicker than in the nonLHON group (Figure 5) in center, indicating that these layers are better preserved in the LHON group.

This difference was statistically significant for the GCC and INL in the central ETDRS circle (Table 2).

In contrast with layers in the central circle, these two layers are thicker in the nonLHON group than in the LHON group (Figure 4), but after applying the Bonferroni–Holm correction for multiple comparisons, only the difference in GCC thickness in the center remained statistically significant (Table 2).

#### 3.3.4. Retinal Layer Thickness Ratios between Middle and Outer ETDRS Rings, and Central Circle

The results, presented in Appendix A, show that the ratio between the middle ring and central circle in GCC and INL thicknesses was significantly lower in the LHON group compared to the nonLHON group. The ratio between the outer ring and central circle was also lower in LHON group for GCC and INL.

#### 3.3.5. Dominant Optic Atrophy Patients

Three DOA patients (Appendix A) were analysed as a separate group. They showed a thicker ETDRS central circle compared to the nonLHON group but thinner than LHON for GCC and INL. The central INL in DOA group was also thicker than in both nonLHON group and controls. This was similar to the findings in LHON group. In the middle and outer ring, the INL layer was thicker in our DOA patients than in LHON, nonLHON group, and controls. We also noticed that our DOA patients had better preservation of inner retinal layers than both study groups.

### 3.4. Peripapillary RNFL (pRNFL) Analysis

Mixed-models analysis was also conducted to compare the mean thickness of peripapillary RNFL layer between LHON and nonLHON groups for each field. The results are shown in Table 3 and Figure 6.

As can be seen, the pRNFL was thinner in LHON group in comparison with the nonLHON group in almost all fields. After applying the Bonferroni–Holm correction for multiple comparisons, only the difference in pRNFL thickness was statistically significant in the temporal superior field (*p* = 0.001). On the other hand, the mean nasal field pRNFL was thicker in the LHON group yet the difference was not significant.

Our data from the longitudinal analysis of the pRNFL thickness showed stabilization or even progression of thinning in all retinal fields (Appendix A), except in the temporal field, where 10 years after the disease onset modest thickening was observed (Figure 7).

## 4. Discussion

The purpose of this study was twofold: first, to assess the segmentation of all retinal layers according to ETDRS rings and fields in the chronic phase of LHON patients, and secondly, to compare the pattern of segmentation in LHON with the pattern of other presumably inherited optic neuropathies where no genetic cause was established. In a clinical setting, diagnosing the type of optic neuropathy based on the clinical phenotype alone is challenging. Therefore, genetic testing is an essential tool for helping with a specific diagnosis. The reported yield of nuclear and mitochondrial DNA sequencing in bilateral optic atrophy is around 20% and includes pathogenic variants in the OPA1, mtDNA genes, or rarely WFS1, MFN2, POLG, ACO2, and DNAJC30 genes [21]. Consequently, simplex cases with a negative family history are more likely to have a negative testing result, even when the clinical presentation is suggestive of LHON.

In a number of cases, the clinical picture may be similar, and after the exclusion of all known acquired causes (infectious, inflammatory, compressive, paraneoplastic, vascular, toxic, etc.), there might be a dilemma whether we are dealing with an atypical mitochondrial pathogenic variant causing LHON or optic neuropathy caused by a yet undiscovered genetic defect that might or might not affect mitochondrial function. Such an example was the autosomal recessive DNAJC30 variant, described recently [2]. Therefore, in this study, we tried to determine whether optic neuropathies due to known mitochondrial dysfunction may have a different pattern of atrophy across the retinal layers in comparison to similar clinical cases without genetic indication of LHON [16]. To the best of our knowledge, such a comparison hasn’t been done yet.

In a comparative segmentation study with healthy controls, both of our patient groups had more than 50% thinner GCC (RNFL-GCL-IPL complex) in all ETDRS fields which is in concordance with previously reported data [10,13,22]. According to Asand and co-workers [23], thinning of the RNFL and RGC-IPL was observed in all 4 quadrants. Zhang et al. [9], on the other hand, reported no difference in the central overall macular thickness in the chronic stage (more than 12 months after the disease onset) in the ETDRS center between controls and their LHON cohort, whilst in all other ETDRS fields the retinal thickness was significantly reduced. In our study, comparable retinal thickness between LHON and the healthy control group in the central circle was also found, whilst in the nonLHON group, it was significantly lower (Table 1). There was also a significant difference observed in GCC in the central circle between LHON and nonLHON group in favor of the former, suggesting that the LHON patients may have better preserved central retina. In contrast to the central circle, the middle and outer ring showed greater thinning of GCC layer in the LHON group. The middle ring corresponds to an area of the parafoveal retina which has 2 or more RGC layers that decline to a single layer when extended to the peripheral zone (outer ETDRS ring) [24]. Liu et al. [25] recently observed that the fovea was thicker and foveal pit depth shallower in unaffected carriers and in LHON patients compared to the normal population. This is also somehow in concordance with the observations in our study and perhaps corresponds with the relative preservation of the GCC (RNFL, GCL, and IPL) in the central ETDRS circle.

Similar to LHON, the preservation of the GCC in the central ETDRS circle was also higher in our DOA patients in comparison to the nonLHON group. Therefore, the impact of mitochondrial dysfunction on retinal ganglion cell apoptosis and axonal degeneration may have a different gradient from the center to the periphery than in other optic neuropathies. Additionally, the ratio in retinal layer thickness between the middle ring and central circle was significantly lower in the LHON group than in the nonLHON group, suggesting the different pattern of retinal degradation in LHON and other types of optic atrophy and possibly providing a new orientational clinical biomarker for differentiation between the two. Likewise, in DOA patients, the middle and outer ring RNFL, GCL, IPL, and INL were also better preserved than in the nonLHON group suggesting better preservation in DOA both in the centre and in the periphery.

In our opinion, this difference between the LHON and DOA is due to a fact that DOA is a slowly progressive disease with a very slow thinning rate [26], therefore at the moment of the recording (roughly five years after the disease onset) the middle ring was still relatively preserved. Also, a distinct pattern of progressive thinning of the GCL was recently reported in both Wolfram syndrome and DOA during the decades of disease duration, in contrast to LHON patients in whom no progression of GCL thinning was observed after the first year of the disease [27].

Another interesting observation in our cohorts was that the INL was thicker in both optic atrophy groups in comparison to the controls. Specifically, LHON patients had thicker INL in all ETDRS fields than the nonLHON group, especially in the central circle, where this difference was statistically significant. Thicker INL in LHON was first described by Carbonelli et al. [22], who suggested that this may be due to macular microcysts, however, Majander et al. [13] found thicker INL also in patients without microcysts. We had only one patient with macular microcysts (Patient 4); therefore, this feature did not seem to be responsible for the difference. It is surprising to see the increased thickness of the INL in the retina of LHON patients. Our results confirm this finding for all ETDRS fields with the most significant difference in the central circle for LHON patients, suggesting that thicker INL might be one of the main characteristics of chronic optic atrophy. Normal INL is composed of approximately 52% cone bipolar cells, 9% rod bipolar cells, 18% amacrine cells, 14% horizontal cells, and 18% Müller cells [28](Rania et al. 2021). Thicker INL may be due to an increase in space in conjunction with the reduction of RNFL and GCL-IPL thicknesses, however, the patterns of GCC and INL thicknesses between our study groups were not the same. LHON group had thicker both GCC and INL in the center. It appears that in LHON a different adaptation process is involved, which leads to better preservation of INL and GCC in the central retina.

In addition, OPL was thicker in the outer ETDRS ring in both optic atrophy groups than in the healthy controls. The mean ONL thickness was also greater in all rings in both groups, but the difference was significant only in the LHON group. Carbonelli et al. [22] and Majander et al. [13] also observed OPL-ONL thickening in LHON, especially if macular microcysts were present. Cesareo and co-workers [29] reported thicker ONL and OPL in the DOA patients without microcysts proposing that this thickening is related to the compensatory behaviour of Müller cells which normally regulate neuronal function by clearing excitotoxins and glutamate, producing neurotrophic factors, and aiding consistent processing of information at the neuronal level. However, this does not explain the differences between LHON and nonLHON groups in our study.

Moving towards outer retinal layers, Lam et al. [30] reported that the photoreceptor outer segment (OS) layer was thicker in LHON patients than in carriers and normal subjects. In our study, differences between groups were not significant. Cesareo et al. [29] reported no statistically significant difference in thickness of the retinal pigment epithelium and outer retinal layers compared to controls.

When comparing pRNFL results in the chronic stage of the disease, we observed thinning of the pRNFL in all segments, especially in the temporal part of the optic disc. The LHON and nonLHON groups significantly differed in temporal and temporal superior fields, whereas the RNFL was thinner in the LHON group. Barboni et al. demonstrated the greatest thickness reduction in temporal and inferior quadrants of the pRNFL in LHON as well as in DOA [26,31].

Asand et al. [23] reported no significant differences in pRNFL thickness between chronic LHON and DOA in the temporal sector, suggesting that these fibers are depleted equally or absolutely. The pRNFL of the inferior and superior field was also severely reduced in both chronic LHON and DOA eyes, although significantly more in LHON, especially in the superior quadrant. They suggested that significant thinning of the superior field pRNFL is a good differential between the two diseases. Wang et al. [32] reported thinning of the pRNFL even at 60 months from onset; in fact, pRNFL thinning continued to occur in some quadrants even more than in the second year of the disease. The thinning was the greatest in the superior quadrant.

The greatest difference in pRNFL thickness in the superior temporal field between LHON and nonLHON group is in concordance with the findings of Wang et al. [32], who suggested that in LHON, sleeping RGC exist that slowly die during the first five years of the disease. We also observed a statistically significant pRNFL thinning in the superior field in LHON in comparison to other optic atrophies, so we think that axons of the sleeping RGCs might project in this part of the optic nerve. The observed thinning might correspond to the previously described slow destruction of those sleeping RGCs and might be the structural biomarker of LHON.

Asand et al. [23] and Barboni et al. [26] reported less significant thinning of the nasal pRNFL in DOA and LHON patients. On the other hand, Barboni et al. [27] recently reported significant progressive pRNFL thinning in all fields in patients with Wolfram syndrome (WS), which was significant in the nasal field, relative to DOA patients which confirms recent theories that WS is not mitochondrial disease [27]. They suggested that relative nasal field pRNFL preservation is characteristic of mitochondrial diseases. Interestingly, our results showed thicker nasal pRNFL in LHON compared with the nonLHON group even though not significantly, suggesting a possibly different pattern in these two optic atrophy groups. Additionally, the longitudinal analysis showed that after 10 years some pRNFL thickening starts in the temporal field only in the LHON group. This pRNFL is partially made from the nerve fibers of the RGC from the central ETDRS circle and is in concordance with the continuous thickening of the RNFL in the central ring which was also observed in our cohort. Continuous thinning of the RNFL in the superior temporal field even after five years of the disease onset, relative preservation of the nasal RNFL, and to some degree recovery of the temporal pRNFL might be characteristics of chronic LHON. A similar observation for mitochondrial optic atrophy patients was made by other researchers. Mashima et al. [33] suggested that the nerve fiber bundles on the temporal side of the fovea might have greater potential to recover their function after the onset of LHON than those on the nasal side. Due to a higher spatial density, the dendritic fields of midget and parasol ganglion cells in the nasal quadrant are smaller than those of cells in the temporal, upper, and lower quadrants. However, the correlation of macular thickness with visual acuity is still not clear. Moster et al. [34] suggested that each micron of GCL thinning was associated with 0.05 logMAR vision loss whilst Zhang et al. [9] described no significant correlation between the BCVA and the macular thickness in LHON patients. Even with significant visual acuity improvement, the average macular thicknesses keep decreasing which indicates that there is no direct relationship with visual function [10] In keeping with this, we found no significant functional difference between the groups either in visual acuity or electrophysiology.

The main limitation of this study is the small number of LHON patients due to the low incidence of this disease in the general population. Similar phenotypical conditions with negative genetic diagnosis despite mtDNA and WES sequencing are rare as well but occur in clinical practice. The obvious question is how to compare a known disease with an unknown one. However, this is a frequent question in a clinical setting, and it would be of help if some biomarkers existed to suggest the diagnosis until the final genetic confirmation is obtained. It is possible that some of the patients in the nonLHON group will be genetically clarified later, as was the case with our patients with DNAJC30 variants. However, there was only one patient (nonLHON 2) in nonLHON group with similar segmentation characteristics to the LHON group. To compare our results with the results of other normative and comparative studies [17,21], we used the ETDRS grid for quantification of the retinal layers although the 1/2.2/3.45 subfield grid may be better suited for ganglion cell representation. However, as the same methodology was used for comparisons, the differences are comparable.

## 5. Conclusions

In our study, we compared chronic LHON patients with those affected by other phenotypically similar primary optic neuropathies in whom comprehensive genetic analysis failed to show causative mutation and present diagnostic challenge. Our findings suggest that LHON patients have relative preservation of the foveal GCC and INL with a trend of pRNFL thickening in the temporal field over time compared to other optic neuropathies. This may provide some support to relative preservation and restructuring in the central retina that may be the basis of potential for visual acuity recovery observed in some LHON patients. It appears that the ratios between the middle ring and the central circle thicknesses for RNFL and GCL closer to 1.5 are indicative for LHON, whilst the ratios over 2 are more suggestive for other optic neuropathies. In clinical practice, our results might be helpful for distinguishing chronic LHON from other types of optic atrophy.

## Figures and Tables

**Figure 1 jcm-11-06045-f001:**
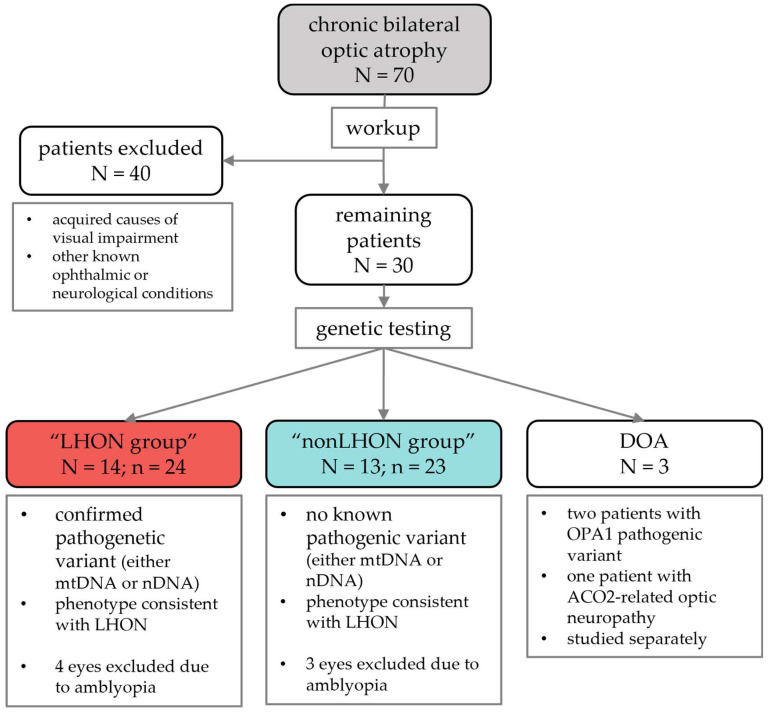
Flowchart of patient management. LHON—Leber hereditary optic neuropathy; DOA—dominant optic atrophy; mtDNA—mitochondrial DNA; nDNA—nuclear DNA; N—number of patients; n—number of eyes.

**Figure 2 jcm-11-06045-f002:**
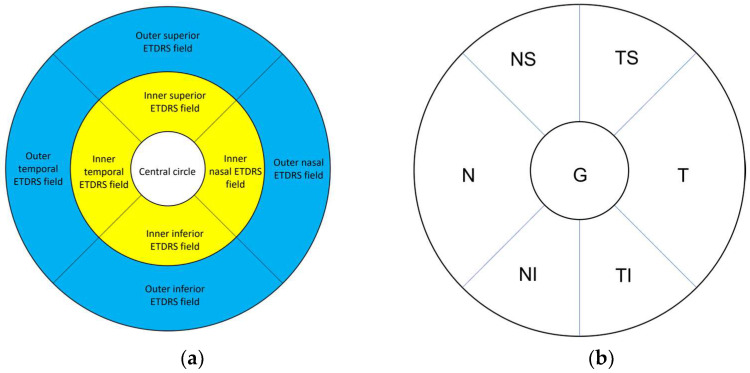
(**a**) ETDRS fields of the right eye. White: central circle. Yellow: Middle ETDRS ring consisting of inner nasal, inner superior, inner inferior, and inner temporal ETDRS field. Blue: Outer ETDRS ring consisting of outer nasal, outer superior, outer inferior, and outer temporal ETDRS field (**b**) Six sectors of the peripapillary RNFL thickness fields of the left eye: temporal superior (TS), temporal inferior (TI), temporal (T), nasal superior (NS), nasal inferior (NI) and nasal (N) fields) and the global 360° average value (G).

**Figure 3 jcm-11-06045-f003:**
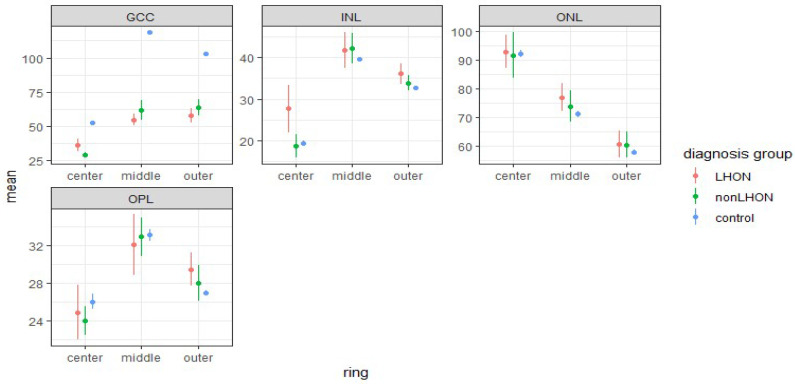
Mean thickness and 95 % confidence intervals of different retinal layers for LHON, nonLHON, and control group, for the center, middle, and outer ETDRS ring. Note thicker inner nuclear layer (INL) in the center of LHON group.

**Figure 4 jcm-11-06045-f004:**
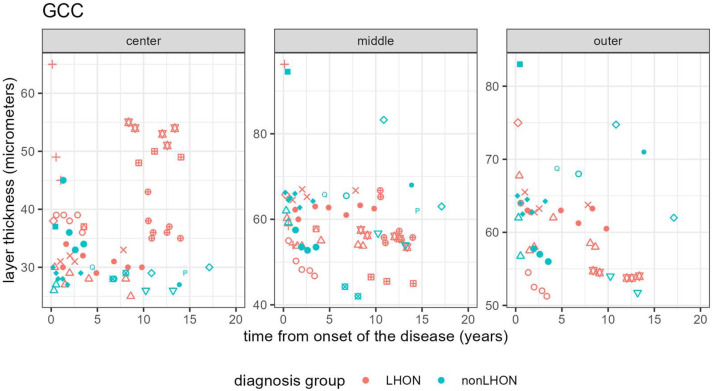
Longitudinal analysis of GCC thickness in ETDRS center, middle and outer ETDRS ring showing a trend of thickening in the center in the LHON group in the late chronic phase of the disease. Different symbols refer to a particular patient, symbols in red belong to the LHON patients, whilst nonLHON patients are marked with green symbols.

**Figure 5 jcm-11-06045-f005:**
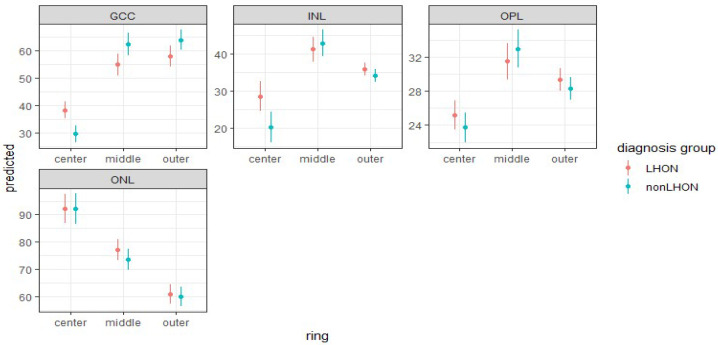
Predicted values for mean thickness (and 95 % confidence intervals) for different retinal layers for LHON and nonLHON group based on full model that includes participants’ ID as a random factor and diagnosis group as a fixed factor. Note thicker GCC layer in the center in LHON group.

**Figure 6 jcm-11-06045-f006:**
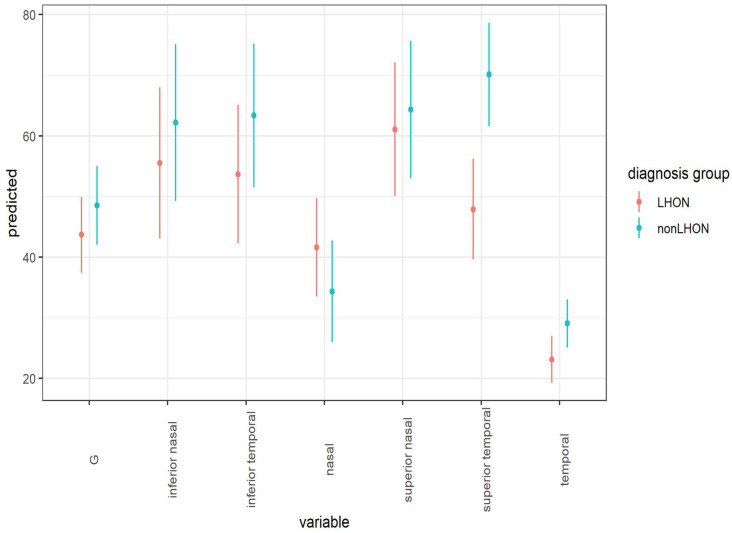
Predicted values for mean thickness (and 95 % confidence intervals) for RNFL layers for LHON and nonLHON group based on full model that includes participant’s ID as a random factor and diagnosis group as a fixed factor. Note thinner RNFL layer in superior temporal quadrant and thicker nasal RNFL in LHON group.

**Figure 7 jcm-11-06045-f007:**
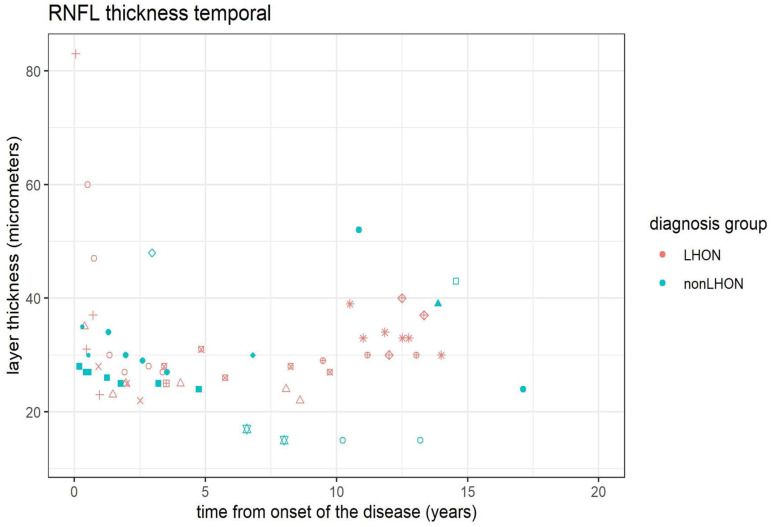
Longitudinal analysis of the pRNFL trend in LHON and nonLHON group shows a trend of thickening of the temporal field pRNFL in the LHON group. Different symbols refer to a particular patient, symbols in red belong to the LHON patients, whilst nonLHON patients are marked with green symbols. Note tendency of thickening in the temporal field in LHON group.

**Table 1 jcm-11-06045-t001:** Comparison of the mean difference in retinal thickness between control group. against LHON and nonLHON group using Welch’s t-test (*p*-values).

	Control vs. LHON (Corrected *p*-Value)			Control vs. nonLHON (Corrected *p*-Value)		
	ETDRS Ring			ETDRS Ring		
Retinal Layer	Center	Middle	Outer	Center	Middle	Outer
Retina	0.101	<0.001	<0.001	<0.001	<0.001	<0.001
GCC	<0.001	<0.001	<0.001	<0.001	<0.001	<0.001
INL	0.234	1	0.275	1	1	1
OPL	1	1	0.245	0.454	0.846	0.257
ONL	1	0.5	1	1	1	1

**Table 2 jcm-11-06045-t002:** Comparison of model fit for the baseline and full model using likelihood ratio test between the LHON and nonLHON group showing the significant differences between LHON and nonLHON group in the thickness of the GCC in the ETDRS center in favor of the LHON group.

	Center			Middle			Outer		
Retinal Layer	c^2^	*p*	Corrected *p*-Value	c^2^	*p*	Corrected *p*-Value	c^2^	*p*	Corrected *p*-Value
GCC	11.92	0.001	0.016	5.95	0.015	0.360	4.34	0.037	0.820
INL	7.3	0.007	0.200	0.45	0.503	1	1.82	0.177	1
OPL	1.4	0.237	1	0.86	0.354	1	1.12	0.289	1
ONL	0	0.988	1	1.64	0.201	1	0.1	0.754	1

**Table 3 jcm-11-06045-t003:** Comparison of model fit for the baseline and full model using likelihood ratio test. The pRNFL was statistically significantly thinner in the temporal and temporal superior field in the LHON group in comparison to the nonLHON group.

Outcome	χ^2^ _(1)_	df	*p*	Corrected *p* Value
RNFL thickness general (G) trend	1.09	1	0.297	1
RNFL thickness inferior nasal trend	0.53	1	0.468	1
RNFL thickness inferior temporal trend	1.29	1	0.257	1
RNFL thickness nasal trend	1.43	1	0.232	1
RNFL thickness superior nasal trend	0.16	1	0.687	1
RNFL thickness superior temporal trend	10.57	1	0.001	0.008
RNFL thickness temporal trend	4.03	1	0.045	0.268

## Data Availability

The data presented in this study are available on request from the corresponding author.

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
