# Peer review of "The Relative Preservation of the Central Retinal Layers in Leber Hereditary Optic Neuropathy"

_jcm, 2022, doi:10.3390/jcm11206045_

Round 1

Reviewer 1 Report

Dear Authors,

I wish to submit my review for the article titled: "Relative preservation of central retinal layers in Leber hereditary optic neuropathy".

The article is well designed and written, and the findings are interesting. In addition, it is clear, comprehensive, and well-written. Tables and images are appropriate. The Authors should be commended for their work.

Could you please upload some OCT images despite the attractive tables and pictures? It may help readers to better focus on your findings.

Page 14: lines 483: Please Correct the typo. "The This may provide "

Author Response

Dear Editor,

We are thankful to the reviewers for the valuable comments, which greatly improved the manuscript.

We have uploaded the manuscript with corrections according to the reviewer’s requests. Track changes have been turned on for  easier review of the changes.

Please find below our point-by-point responses to every reviewers’ comments.

We hope that  our responses with explanations adequately fulfilled all the requests made by the reviewers and that the manuscript will be accepted for publication in your esteemed journal..

Best regards,

Prof Marko Hawlina, MD, FEBO

Responses Reviewer 1

Could you please upload some OCT images despite the attractive tables and pictures? It may help readers to better focus on your findings.

Thank you for this suggestion. New figure with OCTs has been added.  To exemplify the differences between LHON and nonLHON group, we have enlarged the most illustrative segment, however It was still rather difficult to clearly depict differencs in GCC layer that are in the range of several micrometers.

Page 14: lines 483: Please Correct the typo. "The This may provide "

The typo has been corrected

Reviewer 2 Report

In the present work, the authors compared the thickness of retinal layers in the macular region as well as the peripapillary RNFL thickness in LHON in the atrophic stage to presumably inherited optic neuropathies without identified genetic cause in the atrophic stage. The data showed that there are significant difference in RNFL thickness in ETDRS circle and parapapillary region among LHON, nonLHON and control groups, which may represent novel biomarkers as well as a structural basis for possible recovery in some patients with LHON. The study was well designed and analyzed. However, there are several improvements should be made prior to publication

1. A lot of background information was missing from paragraph 2 of Introduction section, the author only briefly described the possible mechanism of causing RGC and papillomacular bundle damage. Please quote more reference to support your statement. Moreover, make this paragraph more closely related to paragraphs 1 and 3.

2. In paragraph 4 of Introduction section, the author declared that nonLHON is special subgroups with typical LHON phenotype but without mitochondrial or nuclear DNA damage. Please quote some reference to support.

3. In 2.1 of materials and methods section, is the patient's diagnostic criteria made by the same person? How to control the accuracy and consistency of your diagnosis?

4. How you calculate the sample size of this study?

5. In 2.1 of materials and methods section, I suggest that the process of patient screening can be represented by a flow chart.

6. The accuracy of genetic test of whole mitochondrial DNA in the included patients was the basis of the present study. So, where was the patient's genetic test done? This information should be added in the manuscript.

7. What is the statistical software used in this study? This should be added in 2.4 sectoin.

Author Response

Dear Editor,

We are thankful to the reviewers for the valuable comments, which greatly improved the manuscript.

We have uploaded the manuscript with corrections according to the reviewer’s requests. Track changes have been turned on for  easier review of the changes.

Please find below our point-by-point responses to every reviewers’ comments.

We hope that  our responses with explanations adequately fulfilled all the requests made by the reviewers and that the manuscript will be accepted for publication in your esteemed journal..

Best regards,

Prof Marko Hawlina, MD, FEBO

Responses Reviewer 2

  1. A lot of background information was missing from paragraph 2 of Introduction section, the author only briefly described the possible mechanism of causing RGC and papillomacular bundle damage. Please quote more reference to support your statement. Moreover, make this paragraph more closely related to paragraphs 1 and 3. The second paragraph of the Introduction section was redone and suggested references were added.

  1. In paragraph 4 of Introduction section, the author declared that nonLHON is special subgroups with typical LHON phenotype but without mitochondrial or nuclear DNA damage. Please quote some references to support. What was meant by our classification in the manuscript was that nonLHON group represent a subgroup of optic atrophy patients in whom, despite all the efforts described in the paper, the cause of the optic disc atrophy could not be identified. These patients might harbor some mutation which has not yet been discovered (as it was a case with recently published arLHON by Stenton et al. 2021) , or the group in which combination of individually nonpathological variants might cause the LHON phenotype (Caporali L et al. 2018) Therefore we corrected the sentence to:“Patients with typical LHON phenotype but without currently discoverable changes neither in mitochondrial nor nuclear DNA are a special optic atrophy subgroup and diagnostic enigma.”

  1. In 2.1 of materials and methods section, is the patient's diagnostic criteria made by the same person? How to control the accuracy and consistency of your diagnosis? The neuroophthalmological team has made the diagnostic criteria (M.H, A.F, M.J.V, and M.S). All patients have been examined/controlled by all the team members and all the team members have agreed upon the diagnosis.

  1. How you calculate the sample size of this study?

 The sample size for this study was not calculated. All patients with bilateral optic atrophy in the population of Slovenia were included in this study. We initially identified 70 patients with chronic bilateral optic atrophy and after the detailed work-up and genetic testing, 40 patients were excluded due to some other identified causes of the bilateral optic atrophy (30 patients remained in the study).

These patients were divided into two study groups according to genetic testing results and phenotype findings. The first group, referred to as “LHON group”, consisted of participants with a confirmed pathogenic variant in either mtDNA or nuclear DNA (nDNA), and phenotype consistent with LHON (14 patients, 24 eyes, mean age 37.5 years, 10 males).

The second group, referred to as the “nonLHON group”, consisted of participants in whom no pathogenic variants were found in either mtDNA or nDNA, but their phenotype included bilateral optic neuropathy without any other detectable cause (13 patients, 23 eyes, mean age 46 years, 7 males).

In three patients, dominant optic atrophy (DOA) was identified and they were studied separately.

  1. In 2.1 of the materials and methods section, I suggest that the process of patient screening can be represented by a flow chart.

Thank you for this suggestion that really clarifies the above mentioned classification. The flow chart of the patient screening has been added to the Materials and methods section

  1. The accuracy of genetic test of whole mitochondrial DNA in the included patients was the basis of the present study. So, where was the patient's genetic test done? This information should be added in the manuscript. All patients were tested at the Clinical Institute of Genomic Medicine, University Medical Centre Ljubljana, Ljubljana, Slovenia which is internationally renowned center for medical genetics. The pertinent geneticists are co-authors of the paper as noted in the manuscript.

  1. What is the statistical software used in this study? This should be added in 2.4 sectoin.

Statistical analysis was performed with R version 3.5.1 (R Foundation, https://www.r-project.org/foundation/)

The data was added to the manuscript.

Round 2

Reviewer 2 Report

The author has modified the article according to the suggestions, and it is recommended to accept it directly